# Influence of deficit irrigation levels on agronomic performance of pepper *(Capsicum annuum* L.) under drip at alage, central rift valley of Ethiopia

Seid Mohammed[1], Arebu Hussen[2]*

1 Department of Irrigation, Alage ATVET College, Zeway, Ethiopia, 2 Department of Plant Science, Mekdela Amba University, Tulu Awulia, Ethiopia

* arebu.hussen@gmail.com

**Data Availability Statement:** All relevant data are within the paper.

**Funding:** Ministry of Agriculture are sponsored to do this research but not include the publication fee.

## Abstract

Water scarcity is one of the most significant constraints on agricultural production in the world, notably in Ethiopia. In the location where this study was conducted, production is only possible once a year. To make the most use of available water, effective water application technologies must be used, and the feasibility of producing crops in water-stressed scenarios must also be researched. In areas of water shortage, deficit irrigation was an essential approach for raising water production and improving water use efficiency. For this purpose, a field experiment was carried out at Alage ATVET College in Ethiopia's Central Rift Valley during the 2019/20 dry season. The regularly grown cash crop pepper was chosen for experimentation under drip irrigation. The study aimed were to investigate the influence of deficit irrigation levels on agronomic performance and water productivity. Seven deficit levels (DI) namely 60, 50, 40, 30, 20, 10 and 0% were laid out in a randomized complete block design with three replications by using drip irrigation. Water application was used in all deficit levels by managing the demand side. Full irrigation produced the maximum plant height, branch number, fruit weight per plant, fruit diameter, fruit length, marketable and total yield. However, at 20% DI levels, stem diameter, flower and fruit number per plant increased. There were only significant variations in total dry yield at 50% and 60% deficiency levels. Marketable yield was significantly different across all deficit levels. It was not possible to determine the water stress threshold level of pepper due to the large variation in yield, but at 30% DI, the yield reduction was about one-quarter of the 0% deficiency level by withholding 33.4% water. CWUE was significantly different at all deficit levels, demonstrating that as stress levels rise, so does CWUE. IWUE exhibited significant difference only at 0 and10% DI. As a result, it is possible to conclude that using at 30% deficit by withholding 33.4% of water can be used to optimize the yield and water productivity of pepper production at Alage and other areas with comparable agro-ecology.

**Competing interests:** All authors declare that they have no conflicts of interest.

# 1. Introduction

## 1.1 Background of the study

Agricultural sectors were the leading sector in Ethiopian economy, 47.7% of the total GDP compared to 13.3% for industry and 39% for services [1]. Even if it is the leading mostly farmers depend on rainfed agriculture, due to the erratic nature of rainfall and frequent drought, there are frequent crop failures resulting in food shortages [2]. Scarce water resources and lack of control over water were pervasive constraints facing a large number of the rural poor worldwide as well as in Ethiopia [3]. Taffa [4] said water from large and medium size rivers that flow through the hills cannot be used for irrigation in Ethiopia, since they cut deep through the area. These scarce water resources may be possible to produce more than one times when deficit irrigation by properly designed, managed and maintained drip irrigation. Applying water at a very slow rate drip irrigation was capable of delivering water to the roots of individual plants as often as desired and at a relatively low cost [5]. Drip irrigation is often the favored method of irrigation, for steep and undulating slopes, porous and shallows soils, fields having widely varying soils, where water is scarce, expensive, and of poor quality.

The growing method and specific agronomic practices can considerably improve the quality and biological value of pepper fruits [6]. Water availability is the major limiting element in plant productivity in arid locations. Pepper is particularly susceptible to water stress, and a lack of water affects fresh fruit yield [7, 8]. The sequence of morphological, biochemical, and physiological changes caused by water stress has a significant impact on plant growth and development [9]. The leaf water potential, leaf and canopy temperature, transpiration rate, and stomatal conductance are all elements that influence water relations. According to Turner *et al*. [10] report under water stress circumstances, there was a considerable decrease in leaf water potential and transpiration rate, which eventually raised leaf and canopy temperature. Reduced moisture availability causes unfavorable changes in photosynthetic pigments, damages photosynthetic machinery, and hinders the operation of critical enzymes [11], resulting in significant losses in plant growth and yield. Optimal irrigation management is essential to maximize both water productivity and pepper fruit quality [12, 13].

Most of the farmers in central rift valley including the study area use traditional furrow irrigations with poor water management and irrigation practice. Shortage of irrigation water, erratic rainfall and short rainy season are problems to produce crops in the study area. The only Djido River, starting from the Silte zone mountainous area for irrigation of the site is highly utilized and used heavily by upper stream users. Therefore, these field experiments were conducted to examine the influence of deficit irrigation levels on the agronomic performance of pepper (*Capsicum annuum* L.) under drip at Alage, central rift valley of Ethiopia. The following specific objectives:

➢ To evaluate the impact of deficit irrigation on growth, yield and water productivity of Pepper

➢ To identify water stress threshold at different water deficit levels on Pepper under drip irrigation methods.

# 2. Materials and methods

## 2.1 Description of the study area

Alage ATVET College is embraced in the central Rift Valley basins which have a semi-arid agro- climatic zone [14]. The college is located in between Jido Kombolcha and Arsi Negele

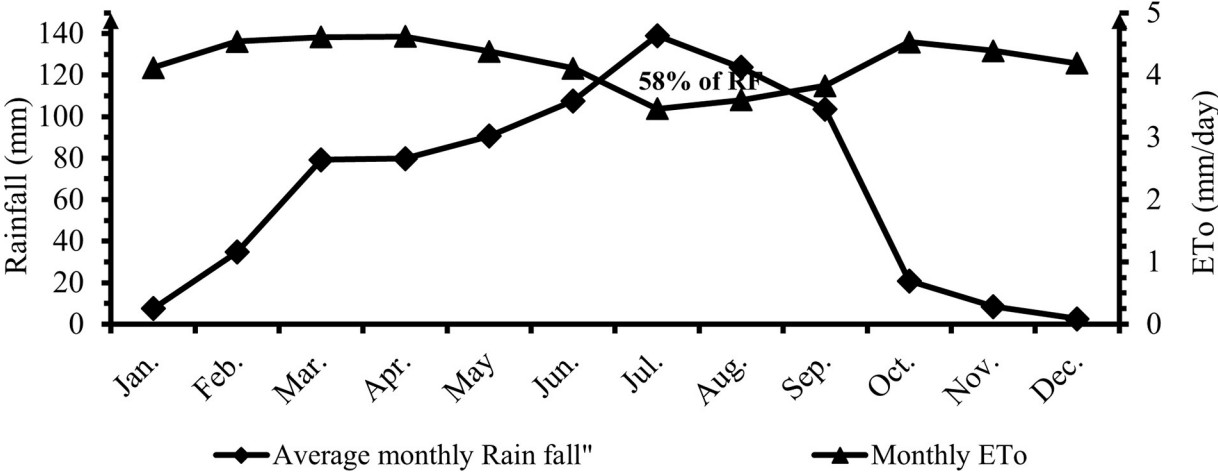

**Fig 1. Long-term rainfall and ET$_O$ of Alage ATVET College.**

Woreda of Oromia regional state and Lanfiro Woreda and Halaba zone of southern Nations and nationalities region. Alage ATVET College is found 214 km far from Addis Ababa with bordering the two Rift valley lakes Abijata and Shalla. It is sited between a geographical coordination's of 70 36' 00"N latitude and 380 24' 55"E longitudes. The average elevation of the college is 1600 m.a.s.l.

## 2.2 Climatic characteristics and soil type

The average annual rainfall is 800 mm. Now a day, the area is experiencing recurrent drought. The annual mean minimum and maximum temperatures were 11 and 29°C [15]. The soil texture of the study area ranges from sandy loam to sandy clay loam with some small areas of loamy and a few clay soils [14]. Long-term rainfall and ET$_O$ of Alage ATVET College was shown in (Fig 1) below.

## 2.3 Experimental field layout and water application

The experimental site was laid out in a randomized complete block design (RCBD) in a factorial arrangement replicated three times per treatment were shown in (Table 1). Treatments consist of seven different deficit levels of water application. Accordingly, the plot size was 2.7 by 2.4 m length and wide with 2 m path between replication containing a total gross area of 312.08 m$^2$. Then the plot area was 6.48 m$^2$ comprising a total of 21 plots were shown (Fig 2). Each experimental plot had four rows of pepper crops nine crops in each lateral line with 36

**Table 1. Treatments setting for the experiment detail layout of the field.**

| | | Treatments | |
|---|---|---|---|
| T$_1$ | Full throughout the growing stage (control) | T$_5$ | 40% deficit throughout the growing stage |
| T$_2$ | 10% deficit throughout the growing stage | T$_6$ | 50% deficit throughout the growing stage |
| T$_3$ | 20% deficit throughout the growing stage | T$_7$ | 60% deficit throughout the growing stage |
| T$_4$ | 30% deficit throughout the growing stage | | |

Note: **T**-irrigation treatment

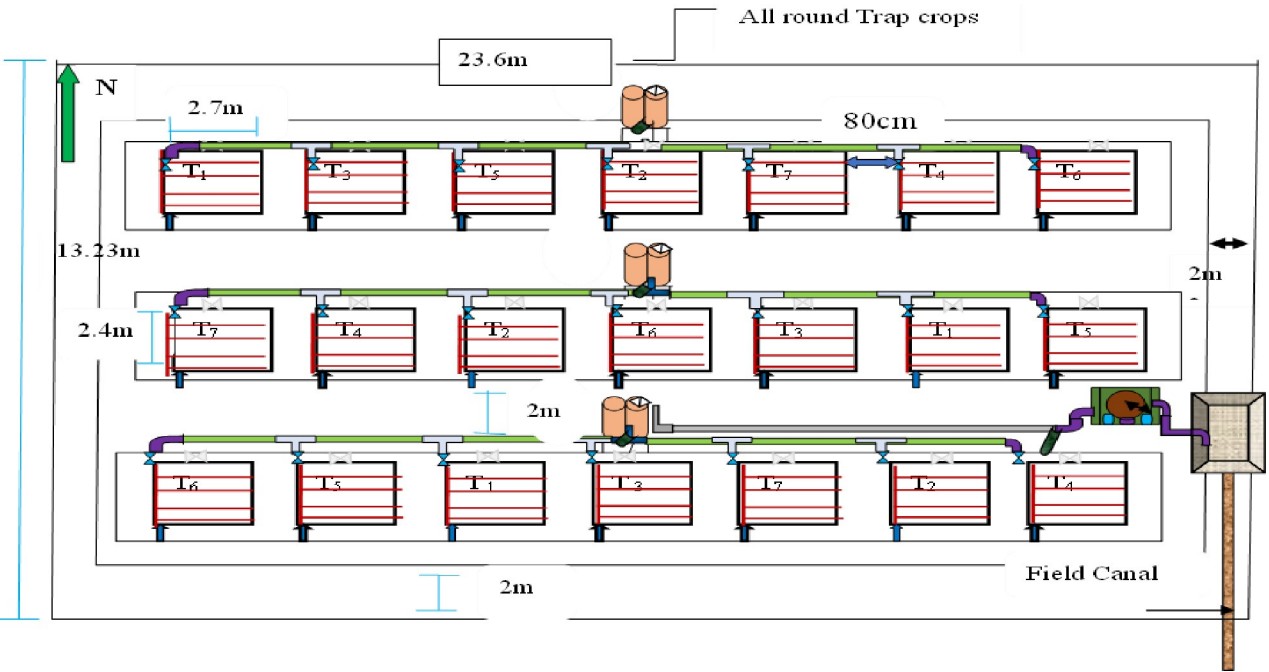

**Fig 2. Layout of experimental field.**

seedlings in each plot. The spacing between the dripper and laterals was 30 and 60 cm respectively.

Water applied to crop was by three-meter length in-line drip with 0.394 lit./hr. discharge and 90.26% emission uniformity. Using paired container (Barrel) with 400 litter volume sets the middle of plot. The main pipe was a high-density pipe (HDP) and ran for 23.6 m length and 25 mm internal diameter for each replication. The water was controlled by an installed butterfly control valve for each plot by taking a water sample from each dripper application time was calculated. The water source was the Djido River starting from the mountainous area of the Silte Zone, and finally drains into Lake Shalla. The total mean annual flow of the river was estimated to be 172 million m$^3$ of water [14]. The water is stored in a trapezoidal shape night storage pond and then pumped by a solar future pump to the tanker.

## 2.4 Mareko fana water requirement estimation and transplanting

Reference evapotranspiration was estimated by the Penman-Monteith method by using cropwat 8.0. The input dates were location (70 36' 00"N latitude and 380 24' 55"E longitudes and altitude of 1600 m), climatic data (sunshine, wind speed, relative humidity, temperature and rainfall), crop type and soil data (OMC, infiltration rate, field capacity, permanent wilting point and total available water). During calibrating the software four days of irrigation intervals and with 90% efficiency.

$$ETo = \frac{0.404(R - G) + \gamma T + \frac{900U_2}{T+273}(e_s - e_a)}{\Delta + r(1 + .34U_z)} \tag{2.1}$$

Where $ET_o$ Reference crop evapotranspiration (mm) Soil heat flux density (MJ/m$^2$d), T-Air temperature at 2m heat (˚c), $U_2^-$ Wind speed at 2m height (m/s).$e_{s-}$ Saturated vapor

pressure ($K_{pa}$), $e_a^-$ actual vapor pressure ($k_{pa}$), Δ- Slope of vapor pressure curve ($K_{pa}$), R-Net radiation at a crop surface ($MJ/m^2d$) and -Psychometric constant ($k_{pa}/°c$).

The ETO was estimated from daily climate data collected from Alage Meteorological Station and computed using CROPWAT 8.0 for windows model and Kc for pepper was taken as 0.6 at initial stage, 0.6 to 1.05 at development stage, 1.05 at mid-season and 0.9 at late season [16].

Then crop water requirement can be computed by empirical methodologies primarily based on meteorological data in view of difficulties associated with direct measurement [17].

$$ETc = Kc*ET_o \tag{2.2}$$

Where, Kc = Crop factor, $ET_o$ = Reference evapotranspiration.

Uniform, healthy, and vigorous seedlings (standard seedlings) having a height of 20–25 cm [18] with 5–8 leaf numbers was transplanted after seven weeks into the experimental site. Full irrigation could apply for one week after transplantation, then after, irrigation was applied as per its deficit levels. The irrigation was demanding base applied every four days interval by adding daily crop water need.

Irrigation water required to meet evapo-transpiration need of crop during its full growth.

$$NIR = ETc - Pe \tag{2.3}$$

Where; NIR net irrigation depth in mm, ETc the crop water requirement in mm and Pe the effective rainfall in mm.

By taking the average application efficiency (90%) for drip irrigation from 85–95% range [19] using Eq 2.4. The gross irrigation could calculate with

$$GIR = \frac{NIR}{Efficiency} \tag{2.4}$$

## 2.5 Soil sampling and analysis

Nine representative soil samples before planting were randomly collected at 0–20, 20–40 and 40–60 cm each depth in a diagonal walk from the experimental field using an auger to have three composite samples. The soil sample was air dried on plastic trays, ground and sieved to pass through 2 mm sieve. Since 90% of pepper root was concentrated at the top 40 cm of soil. Soil Laboratory test was done at Hawassa University, Wendogenet Forestry College. The soil samples were analyzed for physical (texture, bulk density, FC, PWP) and chemical properties (pH, OM, EC and CEC) following standard test procedures. The soil texture was analyzed using Bouyoucos hydrometer method. The textural class of soil was determined using USDA textural triangle [20]. To determine the bulk density (BD), undisturbed core samples were taken using a core sampler from the trial field before planting. Undisturbed sample weighted and oven derided at 105°C for 24 hrs. [20]. The FC and PWP were determined using the pressure plate and membrane apparatus by applying a pressure of 1/3 bar (FC) and 15 bars (PWP), respectively were exerted until no further change in soil moisture content was observed. The total available water (TAW) for plant use in the root zone was computed as the difference in moisture content between field capacity and permanent wilting point at 60 cm depth.

The soil pH was determined by taking saturated extract methods using a pH meter. The organic carbon (%) was determined following the wet digestion method as described by Black [20]. OM content was then calculated by multiplying OC by 1.724. The EC was determined by measuring the conductivity of saturated soil extract using an electrical conductivity meter at 25°C. Infiltration capacity of experimental field was determined using a double ring infiltrometer on three replications. It was measured before the experimental work.

## 2.6 Hydraulic characteristics of drip irrigation system

Drip emitters flow in the experiment was measured and checked by taking water samples from three levels of water at the tanker (full, half and minimum level) and averaging it. Three catch cans were randomly assigned each plot was placed beneath drip emitters by excavated the soil. Water dropout from each emitter was collected for five minutes and measured using a graduated cylinder. Emitter discharge was determined by dividing the volume of water collected to time. Hydraulic characteristics of systems were evaluated emitter flow rate, flow rate variation, uniformity coefficient and emission uniformity were data from emitter discharge observations, the emitter's uniformity parameters, were calculated using the following equation [21].

$$qv = \frac{(qmax - qmin)}{qmax} * 10 \qquad (2.5)$$

Where: $qv$ = emitter flow variation (%), $q_{max}$ = maximum emitter flow rate (l/hr.). $q_{min}$ = minimum emitter flow rate (l/hr.).

$$Cv = \frac{S}{qa} \qquad (2.6)$$

Where: CV = Coefficient of variation, S = standard deviation of emitter flow rate and qa = mean emitter flow rate (lit/hr.).

Emission uniformity was measure of uniformity of emitter discharge from all the emitter of drip irrigation system and was the single most important parameter for evaluating system performance. The recommended classification of emission uniformity and design standards for emission uniformity for arid areas were shown in (Tables 2 & 3) below. It shows the relationship between minimum and average emitter discharge [22].

Emission uniformity of the emitter was calculated by the equation given below:

$$Eu = \frac{\text{Average low quarter minimum discharge of emitter}}{\text{over all avarage rate of discharge}} * 100 \qquad (2.7)$$

The length of irrigation time interval could be determined:

$$\text{Irrigation time (hr)} = \frac{\text{water requirement}}{\text{emmiter rate of application}} \qquad (2.8)$$

Irrigation duration calculated using the following formula.

$$\text{Duration of irrigation} = \frac{\text{dripper discharge (lph)}}{\text{dripper spacing} * \text{inline spacing (m)}} \qquad (2.9)$$

**Table 2. Recommended classifications of emission uniformity.**

| EU range | Ratings |
| --- | --- |
| 90% or greater | Excellent |
| 80–90% | Good |
| 70–80% | Fair |
| Less than 70% | Poor |

Source: Gireesh [23]

**Table 3. Design standards for emission uniformity for arid areas.**

| Emitter type | Crop spacing | Field topography | Emission uniformity (%) |
|---|---|---|---|
| Point source | Wide[a] | Uniform[c] | 90–95 |
| | | Steep[d] or undulating | 85–90 |
| | Close[b] | Uniform | 85–90 |
| | | Steep or undulating | 80–90 |
| Line source | Close | Uniform | 80–90 |
| | | Steep or undulating | 75–85 |

Source: Bralts [24].

*Note*: *a* space greater than 4m apart, *b* spaced less than 2m apart, *c* slope less than 2%, *d* slope greater than 2%.

## 2.7 Data collection and analysis

Phenology and growth parameters: Plant phenological parameters such as days to 50% flowering and 50% maturity were recorded on a plot basis from the two central rows. Plant height, number of primary branches per plant, number of leaves per plant, number of flowers per plant, stem diameter, number of pods per plant, pod length, diameter of the fruit and fruits weight per plant were recorded on plants basis by selecting five plants randomly from each plot. At maturity, whole plants from the two central rows of the net plot area (1.3 m x 2.4 m = 3.12 m$^2$) were manually harvested and sundried to marketable yield, unmarketable yield, total yield, and crop and irrigation water use efficiency.

The yield response factor (Ky) of pepper which relates relative yield decrease to relative ET deficit was estimated using Equation.

$$1 - \frac{Ya}{Ym} = Ky\left(1 - \left(\frac{ETa}{ETm}\right)\right) OR\ Ky\frac{\left(1 - \frac{Ya}{Ym}\right)}{\left(1 - \frac{ETa}{ETm}\right)} \tag{2.10}$$

Where, $Y_a$ and $Y_m$ are actual and maximum crop yields, corresponding to $ET_a$ and $ET_m$ actual and maximum evapo-transpiration, respectively; $K_y$ was yield response factor.

The collected data was subjected to analysis of variances (ANOVA) using GenStat 15[th] [25] and significant treatment means were compared using least significant difference (LSD) at P<0.05 probability level.

## 3. Results and discussion

Infiltration rate and cumulative infiltration of the experimental site were 5.1 and 22.46 cm/hr. respectively. The basic infiltration rate of sandy loam soil is within the range of 1.3 to 7.6 cm/hr. [26] with a higher infiltration rate which was the typical characteristic of sandy loam soil. The experimental site soil pysico-chemical properties were shown below on (Table 4).

## 3.1 Monthly reference evapotranspiration (ETO)

The long term as well as in the research time reference evapo transpiration (ET$_O$) calculated from 2009 to 2019 value of the area was presented in (Fig 3).

The long-term ET$_O$ ranged between 3.6 mm/day in July to 4.6 mm/day in March. In research time as well as in long term evapotranspiration March has the maximum reference evapotranspiration. The overall average of 3.84 mm/day for the whole growth period at the research time. Applied net irrigation water depth at different deficit Levels, based on climatic data of Alage, the net seasonal irrigation water applied was under here.

**Table 4. Soil physico-chemical properties of the experimental site.**

| Soil property | | Soil depth in (cm) | | | |
|---|---|---|---|---|---|
| | | (0–20) | (20–40) | (40–60) | Average |
| Soil Texture | Sand | 58 | 64 | 64 | 62 |
| | Silt | 28 | 24 | 22 | 25 |
| | Clay | 14 | 12 | 14 | 13 |
| | Textural Classes | Sandy loam | Sandy loam | Sandy loam | Sandy loam |
| Bulk density (gm/cm$^3$) | | 1.4 | 1.6 | 1.6 | 1.53 |
| FC (%) | | 20 | 20.2 | 20.8 | 20.33 |
| PWP (%) | | 10.5 | 9.7 | 10.5 | 10.2 |
| TAW (mm/0.6m) | | 31.4 | 33.6 | 33.0 | 98.0 (total) |
| PH | | 7.69 | 8.1 | 7.71 | 7.83 |
| ECe $\left(\frac{\mu s}{cm}\right)$ 25˚C | | 60.7 | 68.7 | 92.2 | 73.87 |
| Organic Matter (%) | | 0.96 | 0.34 | 0.24 | 0.51 |
| Organic Carbon (%) | | 0.56 | 0.2 | 0.14 | 0.3 |
| Nitrogen (mg/kg) | | 0.048 | 0.0168 | 0.012 | 0.026 |

On the above (Fig 4) net water applied for full irrigation was estimated to be 485 mm. Accordingly, for 90, 80, 70, 60, 50 and 40% ETc level was 425, 388, 340, 291, 243 and 194 mm, respectively with an average irrigation efficiency of 90%. The same result gets by Yibekal [27] within the range of Melkasa and Ziway gross irrigation application.

The result specifies that the maximum amount of water was applied around the end of February up to the second week of March. This matches with the stage of fruit formation (3rd stage) of the pepper. During 1st, 2nd and at last stage low evapotranspiration was observed. This time was characterized by mean values of daily ETc of about 2.8 and 4.7mm at 1st stage and at last stage reference water requirement and crop factor were smaller. The time of crop canopy grew, ETc increased and reached the highest mean value at mid-stage (5.4 mm). Therefore, comparing the ET values for the individual growth periods, it was high at the 2nd and 3rd growth stages than that of the 1st and 4th stages (Table 5). A similar finding was got by Gireesh [23]. This might be due to the higher growth of crops and hot climatic conditions at the research time of these stages.

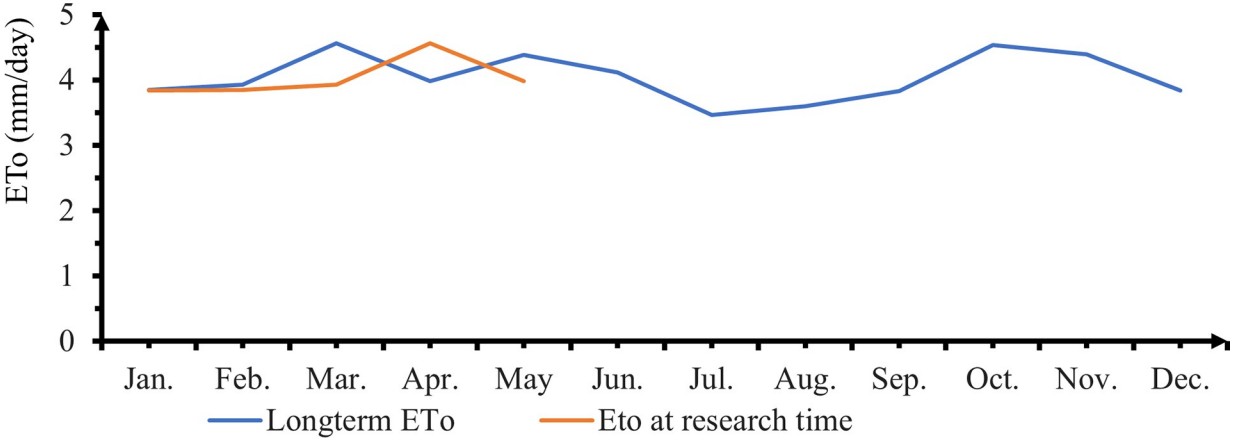

**Fig 3. Long term and research time monthly reference evapotranspiration of the research area.**

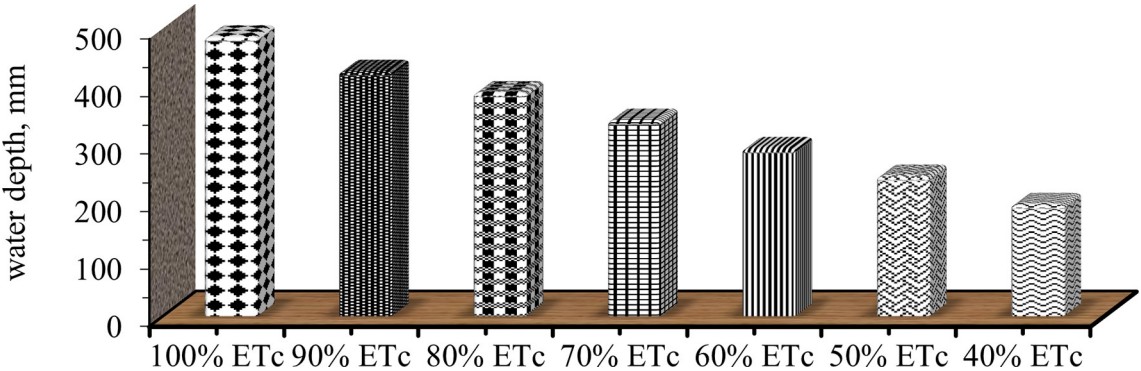

**Fig 4. Applied net irrigation water depth at different deficit levels for mareko fana pepper.**

**3.2.1 Plant height.**   Pepper height showed a highly significant (P<0.01) among the different irrigation levels was presented in (Fig 5) below. A significant difference was observed at 0% and 10, 30 and 60% deficit levels but 10, 20 and 30% showed insignificant differences.

The effect of irrigation levels on plant height treated with 0% DI had the tallest plant height (50.47cm) and was significantly different from all deficit irrigations. The shortest plant height (30.67 cm) was recorded from DI of 60% DI application and was significantly inferior to all irrigation levels. This decrease in growth characteristics might be attributed to a decrease in water flow from the xylem to the various cells, which governs cell division, elongation, and development, as well as a decrease in chlorophyll content and lipid peroxidation in the cell membrane. Anjum *et al.* [28] reported that a deficit of water induced significant alterations in the activity of antioxidant enzymes. Furthermore, peroxidase and catalase activity were predicted to have risen or stabilized during the early water shortage and subsequently decreased as the water deficit persisted. Similar results with Arebu *et al.* [29] and Adel *et al.* [30] confirm when increasing deficit levels exhibited a reduction in vegetative growth, fruit parameter on yield with insignificant increased irrigation water use efficiency and consistent reduction in the amount of irrigation water. In addition, plant height was influenced by water application depth at the development and mid-growth stages [31]. The result was on the contrary to the result of Alvano *et al.* [32] said that the plant height did not differ under the various irrigation levels.

**3.2.2 Leaf number per plant.**   Leaf numbers per plant was significantly (P<0.01) affected by irrigation levels. The effects of deficit irrigation levels insignificant effect on a number of leaves per plant at 0, 10, 20 and 30% DI but the difference started from 40% deficit levels (Table 6). The highest number of leaves per plant (228.30) was observed from 0% DI due to generating tender shoot, and mix soil nutrients ready to receive by plant. Number of leaves per plant was reduced with increased irrigation deficit levels. The lowest number of leaves per

**Table 5. Depth of water applied for each stage of mareko fana pepper.**

| Growth period | Depth (mm) | | | | | | |
|---|---|---|---|---|---|---|---|
| | 100% | 90% | 80% | 70% | 60% | 50% | 40% |
| 1st stage | 82.6 | 74.5 | 66.5 | 57.8 | 50 | 41.42 | 33.1 |
| 2nd stage | 139.7 | 125.8 | 110.8 | 97.7 | 84 | 69.94 | 56 |
| 3rd stage | 234.6 | 211 | 191.7 | 164.1 | 140 | 117.57 | 94.11 |
| 4th stage | 82.1 | 73.7 | 65 | 57.4 | 49 | 41.07 | 32.79 |
| **Total** | **539** | **485** | **434** | **377** | **323** | **270** | **216** |

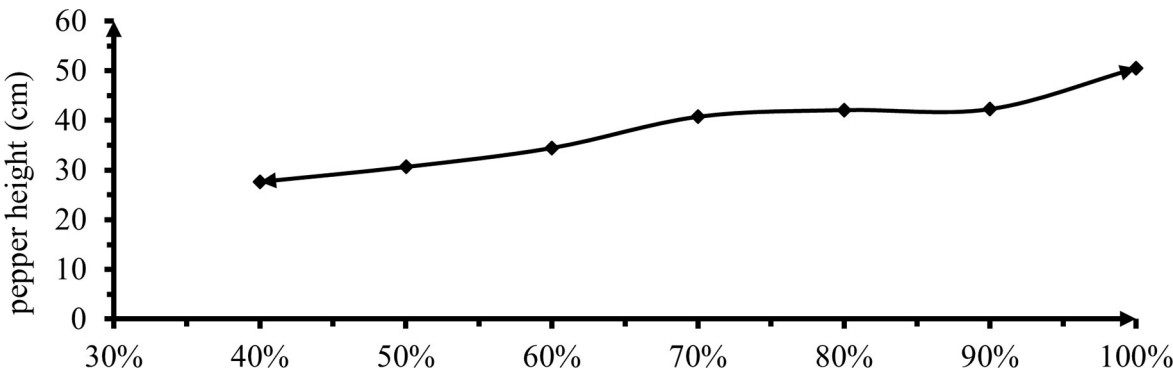

**Fig 5. The effect of irrigation levels on height of mareko fana pepper.**

plant (104.7) was observed from 60% DI and significantly lowest from all irrigation levels as a consequence of water stress on cell expansion.

This result was similar to Yemane *et al.* [33] finding, who reported that the highest leaf number of onion resulted from the application of 100% ETc irrigation depth due to the effect that facilitates nutrient availability and photosynthesis for undisrupted growth of plant. This indicated that plants respond to water stress by closing their stomata to slow down water loss by transpiration, gas exchange within the leaf was limited, consequently, photosynthesis and growth slow down. The leaf expansion is generally determined by the turgor pressure and the availability of assimilates. The result was also agreed with Adel *et al.* [30] found that the highest number of leaves per plant was recorded from control irrigation (100% ETc) and significantly different from all other deficit irrigation levels. Full irrigation has the highest biomass yield followed by moisture stress level of 85% crop water requirement was not statistically different [31]. In general, the result indicated that when irrigation water amount increases the number of leaves per plant also increased.

**3.2.3 Stem diameter.** The obtained result showed that the stem diameter was significantly (P<0.01) influenced by irrigation levels. Stem diameter at 0, 10, 20 and 30% deficit levels show an insignificant difference but a significant difference was observed after 40% DI level (P<0.01). Comparing with effects of different irrigation levels, the maximum stem diameter (1.0021cm) got from 20% DI. This showed, the crop got maximum irrigation levels at 10 and

**Table 6. Analysis of variance of selected agronomic and phonological related parameters.**

| Treatment | PH (cm) | SD (cm) | D50%F | D50%M | LN | NF | BN |
|---|---|---|---|---|---|---|---|
| 100% ET$_C$ | 50.47$^e$ | 0.91$^{cd}$ | 92.6$^e$ | 134.7$^e$ | 228.0$^c$ | 75.47$^c$ | 10.90$^c$ |
| 90% ET$_C$ | 42.27$^d$ | 0.88$^{bcd}$ | 89.60$^{de}$ | 133.1$^{de}$ | 203.7$^c$ | 81.53$^{cd}$ | 10.73$^c$ |
| 80% ETC | 42.07$^d$ | 1.00$^d$ | 85.67$^{de}$ | 131.3$^{cd}$ | 202.7$^c$ | 96.07$^d$ | 9.83$^{bc}$ |
| 70% ET$_C$ | 40.73$^{cd}$ | 0.96$^d$ | 82.47$^{cd}$ | 129.8$^c$ | 215.1$^c$ | 74.43$^c$ | 9.07$^{bc}$ |
| 60% ETC | 34.47$^{bc}$ | 0.81$^{bc}$ | 74.20$^{bc}$ | 126.9$^b$ | 157.0$^b$ | 51.07$^b$ | 7.67$^{ab}$ |
| 50% ETC | 30.67$^{ab}$ | 0.77$^b$ | 68.13$^{ab}$ | 125.5$^b$ | 137.9$^{ab}$ | 36.20$^{ab}$ | 6.60$^a$ |
| 40% ET$_C$ | 27.67$^a$ | 0.64$^a$ | 63.20$^a$ | 117.4$^a$ | 104.7$^a$ | 33.17$^a$ | 5.80$^a$ |
| s.e. | 2.2.2 | 0.042 | 1.95 | 0.737 | 19.71 | 9.57 | 0.79 |
| cv. | 7.1 | 6.1 | 3.0 | 0.7 | 11.0 | 15.0 | 11.2 |
| L.S.D (1%). | 6.78 | 0.129 | 5.964 | 2.252 | 35.07 | 17.02 | 1.724 |

Where: -PH—Plant height, *SD*-Stem diameter, *D50%F*- Days to 50% flowering, *D50%M*- Days to 50% Maturity *DM*-Days to maturity, *CV*-coefficient of variation and *s. e.*-standard error.

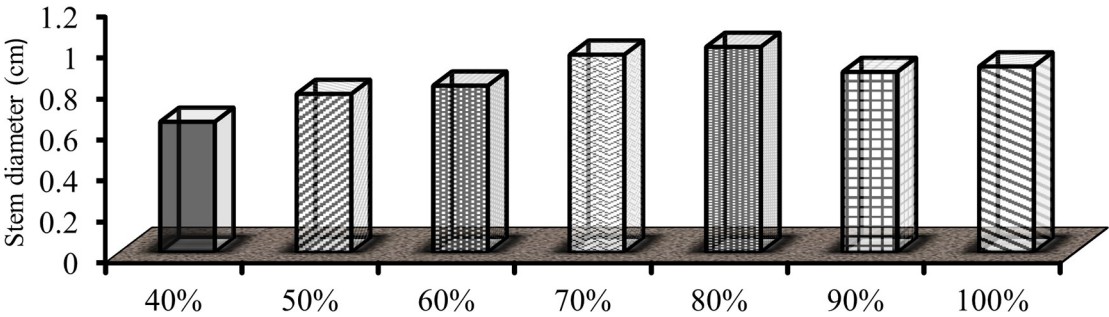

**Fig 6. Effect of irrigation levels on stem diameter.**

0% deficit levels pepper produces a large number of leaves and branches due to this it caused compact to pass enough sunlight and the computation of nutrients resulted from smaller stem diameter than 20% DI. When the deficit level increases vegetative part of the pepper becomes lower consequently resulted minimum stem diameter. Minimum stem girth (0.64cm) was observed from 40% DI with significantly different from other treatments. The reduction in anatomical characters under drought stress may be due to the harmful effect of water deficit on cell division and expansion as well as nutrient uptake [34]. According to Okunlola *et al.* [35], moderate and severe drought reduced the carotenoid, chlorophyll a, b, and total chlorophyll content of the study plants during the vegetative stage. The result was contrary to Padron *et al.* [36], who said that the irrigation level does not affect fruit length and stem diameter, corroborating the results of those who cultivated bell pepper under different irrigation levels, with daily irrigation. Gireesh [23]; Adel *et al.* [30] had conveyed similar results (Fig 6).

**3.2.4 Number of primary branches per plant.** The analysis of variance on effects of irrigation levels for number of primary branches per plant is presented in (Table 6 & Fig 7) below. Number of primary branches per plant at 0, 10, 20 and 30% deficit levels there was a trivial difference. The significant difference was observed after 40% DI level (P< 0.01).

Number of primary branches per plant varied from 5.8 to 10.9 under irrigation deficit levels between 60 and 0% DI. This difference was observed when irrigation levels increased the plant generate more branches and leaf resulted a large pod size. This finding was confirmed by Adel *et al.* [30], the highest number of branches per plant was recorded from treatment that received 100% ETC. On the other hand, the lowest number of branches per plant was recorded from a plot treated with 55% ETC. Similar findings by [37, 38].

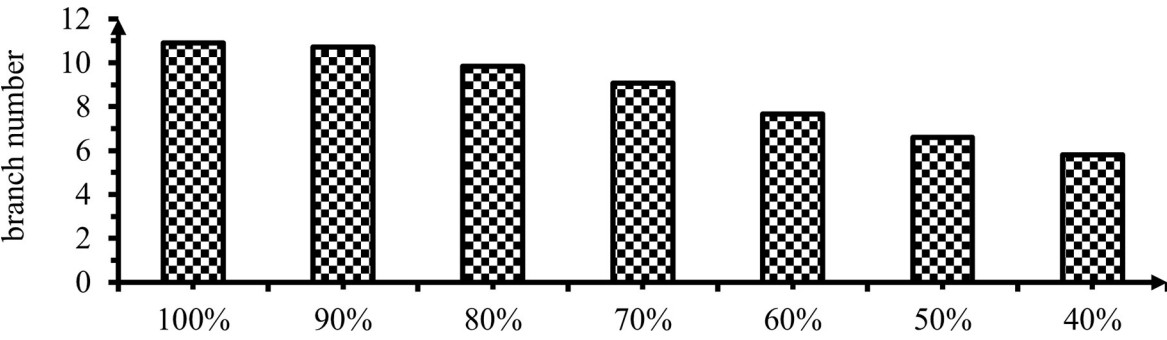

**Fig 7. Effect of irrigation levels on branch number.**

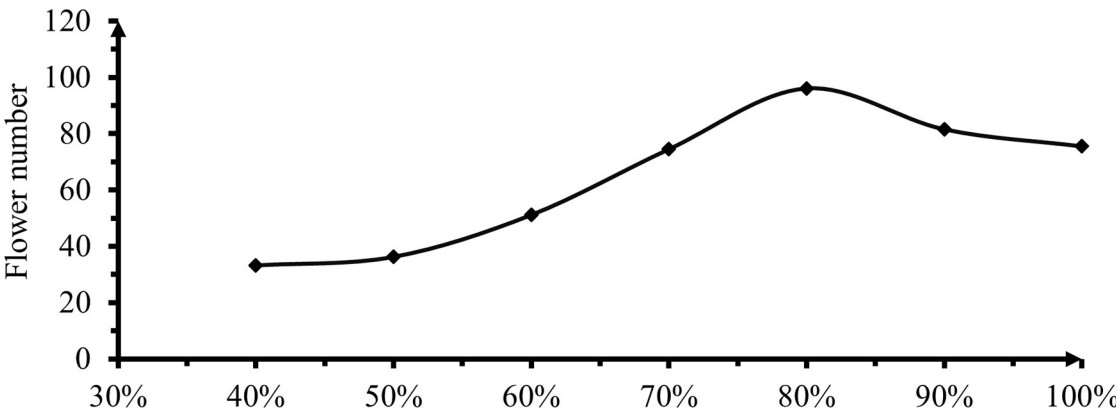

**Fig 8. The effect of irrigation levels on number of flowers per plant.**

**3.2.5 Number of flowers per plant.** The effect of deficit irrigation on number of flowers showed highly significant (P<0.01) in this study showed in (Table 6 & Fig 8). Numbers of flowers per plant at 0, 10 and 30% deficit levels insignificant difference was observed. A significant difference was detected after the 40% DI level (P<0.01) with maximum significant difference levels on the 20% deficit in graph 8.

Accordingly, the highest number of flowers per plant was recorded from 20% deficit level was 96.07 with the same significant levels of 10% DI. The least number of flowers per plant was also observed from 40% ETC level got 33.17 due to water stress to make ready nutrients for the plant. Variations of flowers caused by the effect of irrigation on flowering, lack of optimum soil moisture at flowering stage of crop especially 60% deficit levels and due to higher irrigation amount got at 10 and 0% deficit levels number of flowers become decline.

The result indicated insignificant difference between 60 and 50% and also 20 and 10% deficit levels (Fig 8). Moreover, the primary cause of poor flowering and fruit set as well as marketable yield loss could be due to limited amount of moisture that results in loss of potential fruit, frost causes flower and fruit damage (flower failed) and loss of yield. The inhibitory effects of water stress at flowering and reproductive stage had produced significantly lower yield as compared to optimal irrigation. When water stress occurs at the reproductive stage especially flowering and pod formation, affects the yield more severely than it occurs at other stages. Similarly, Adel *et al*. [30] alleged that deficit irrigation caused a significant hastening in the time of flower set and first harvesting of hot pepper. Also, Arebu *et al*. [29] showed that the effect of a high deficit level reduces a total number of pod production by a failure of flower due to moisture stress during the flowering period. Rainfed plants had lower dry-mass buildup in above-ground plant organs, gas-exchange properties, leaf water potentials, and intercepted solar radiation than irrigated plants. Leaf ion and protein concentrations were higher in rainfed than in irrigated plants [28].

**3.2.6 Days to 50% flowering.** The number of days to 50% flowering revealed a significant (P<0.01) difference in response to the effects of deficit irrigation levels (Fig 9).

Table 6 and Fig 9 showed, 0, 10 and 20% DI had insignificant difference on days to 50% flowering period (P<0.01). Significant differences started at 30% deficit levels. Longer days to flowering 92.6 days were shown in plots with 0% DI while the shortest days to flowering 63.3 days were recorded from treatment received 60% DI. This indicated that when stress levels increased the pepper respond to water stress by ending its life by offered flowers in a short time. The longest flowering period in treatment received 0% DI during the growth period due to effects of a large amount of water application than others, this indorsed vegetative growth

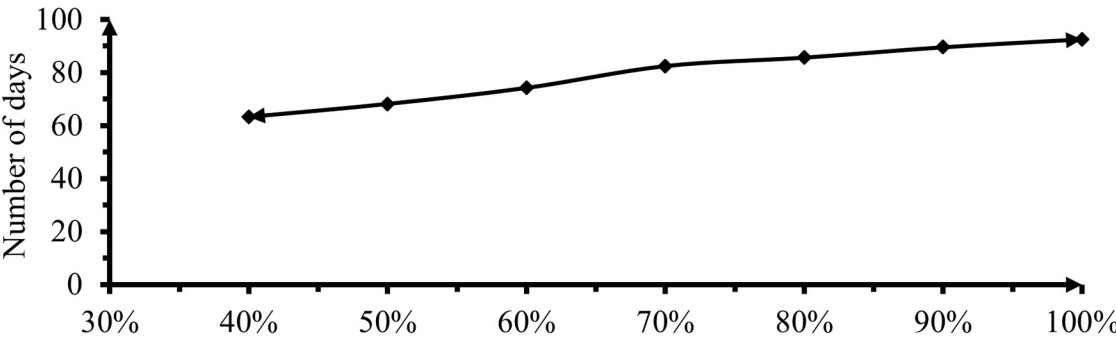

**Fig 9. Effect of irrigation levels on 50% flowering date.**

and delayed transition to the reproductive period. The same result was obtained by Lelisa *et al.* [38] and Ahmed [39].

**3.2.7 Days to 50% maturity.** Number of days to 50% maturity was a highly significant difference in response to the effect of deficit irrigation levels (P< 0.01). 0% and 10 and 20% deficit had insignificant differences on 50% maturity days as shown below in Fig 10. Significant differences started at 30% deficit levels.

Treatment that received 0% DI irrigation required 134.7 days with the same significant level of 10% deficit (Fig 10). Whereas, treatment received 60% DI entails the shortest 117.4 days to 50% maturity and is significantly different from others. As indicated earlier, treatments that received 60% DI set flowers earlier and mature early than treatments that received more irrigation. These results signposted that full level irrigation application delayed phenological periods of pepper. This could be due to promote vegetative growth and extended transition to maturity. This finding disagrees with Gonzalez *et al.* [40] stated that the water deficit did not hasten ripening but reduced biomass production of pepper. Nevertheless, it agreed with Adel *et al.* [30] and Habtie [41] got irrigation deficit levels increased pepper flowering and maturity time become reduced. Yetagesu *et al.* [42] also reported on onion the plant receiving 100% of ETc irrigation had significantly longer days to maturity, however, plants receiving 40% of ETc irrigation level the shorter days to maturity.

ANOVA analysis showed that water levels received an insignificant difference between 50 and 60% irrigation levels. Thus, plants grown with soil moisture deficit levels on average days took 12.84, 6.83, 5.79, 3.64, 2.52 and 2.67% more duration to reach 50% maturity compare

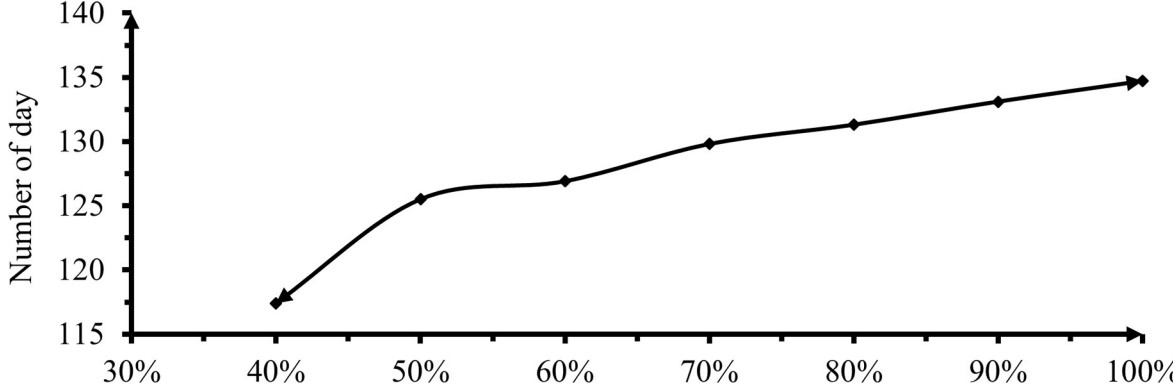

**Fig 10. The effect of irrigation levels on 50% maturity date.**

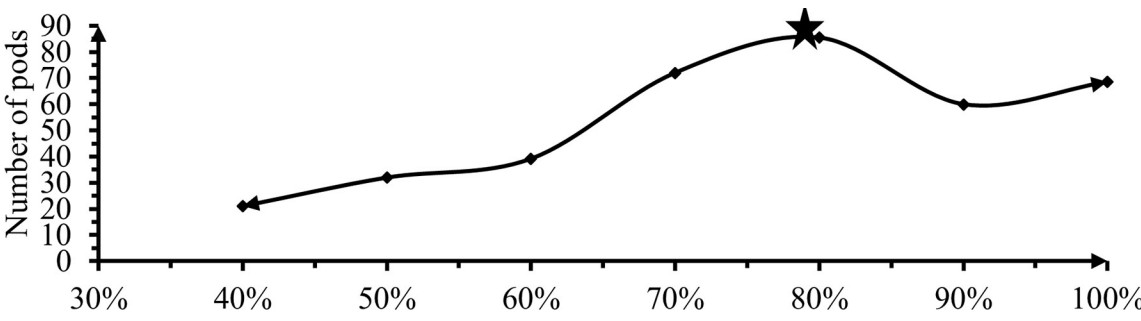

**Fig 11. The effect of irrigation levels on number of pods per plant.**

with full irrigation. This could be because plants under stress are more likely to complete their life cycle in a shorter period of time, allowing them to escape from unfavorable conditions by ending their life cycle a few days earlier than those under normal or high soil moisture conditions, ensuring the species' survival.

### 3.3 Yield and yield components

**3.3.1 Number of pod per plant.** Analysis of variance from ANOVA table for total number of fruits per plant revealed that significant effect (P<0.01) between 60, 50 and 20% DI due to effect of deficit irrigation levels shown in Fig 11.

Maximum number of fruits per plant (85.4) was recorded from 20% deficit irrigation level while minimum fruit number per plant (20.93) was observed from 60% DL which was four times lower than 80% ETC. Insignificant difference between plots that received 30, 20, 10 and 0% DI while significant difference with other levels was shown. Irrigation levels received 20% ETC got maximum flower number with the same sequence levels of 30, 10 and 0% DL (Table 6). These results indicated that differential response of Mareko fana to different deficit levels production of fruits per plant was higher in 20% deficit levels as well as increase irrigation levels also increase pod per plant (Fig 11). Irrigation water deficit may have a negative impact on the quantity of flowers and pods due to a decrease in relative water content, total chlorophyll content, and photosynthetic efficiency, as well as reduced translocation and ion absorption. This result was agreed with the finding of Adel *et al.* [30] and Elias *et al.* [43] as deficit levels increased, number of fruits per plant also increased up to some level of deficit but, after instance decreased.

**3.3.2 Pod length.** Analysis of variance showed, there was insignificant difference (P<0.01) effect on mean fruit length at 0% and 10, 20 and 30% DI and also 50 and 60% DI due to effect of deficit irrigation levels.

The longest fruits (11.53 cm) were obtained from plots received 0% DI while treatment received 60% DI (5.51cm) gave the shortest fruit length compared to other levels (Table 7 & Fig 12). The result was in line with Adel *et al.* [7] said, that increasing deficit irrigation caused significant reduction in fruit length where100% gave the largest fruit length followed by 85, 70 and 55% respectively. Metin *et al.* [44] also confirm that a uniform supply of soil water throughout the growing season is needed to prevent poor fruit size and shape and to increase yield. Contrary verdict with Padron *et al.* [36], the significant difference observed in fruit length was reflected by variation in depth of water applied in respective deficit irrigation levels. The smallest fruit length of 5.51cm in 60% DI levels indicated that a slight imbalance water supply caused a drastic reduction in fruit growth.

**Table 7. Analysis of variance on yield component plant of mareko fana pepper.**

| Treatment | NPP | PL (cm) | PDF (cm) | FWPP (gm) |
|---|---|---|---|---|
| 100% $ET_C$ | 68.47[cd] | 11.53[e] | 2.136[e] | 745.8[f] |
| 90% $ET_C$ | 59.93[bcd] | 9.896[d] | 1.856[de] | 642.1[e] |
| 80% $ET_C$ | 85.40[d] | 9.249[d] | 1.845[cde] | 495.5[d] |
| 70% $ET_C$ | 71.93[cd] | 8.924[cd] | 1.613[bcd] | 259.1[c] |
| 60% $ET_C$ | 39.07[abc] | 7.592[bc] | 1.473[abc] | 205.5[bc] |
| 50% $ET_C$ | 31.87[ab] | 6.440[ab] | 1.237[ab] | 132.9[ab] |
| 40% $ET_C$ | 20.93[a] | 5.511[a] | 1.229[a] | 106.5[a] |
| s.e. | 11.14 | 0.511 | 0.1237 | 29.8 |
| cv. | 25.3 | 7.4 | 9.3 | 9.9 |
| L.S.D. (1%) | 34.01 | 1.56 | 0.378 | 1.03 |

Where: NPP- number of pods per plant, PL- Pod length, PDF -diameter of the fruit, FWPP- fruits weight per plant. CV-coefficient of variation and s.e -standard error.

**3.3.3 Diameter of fruit.** Fruit diameter was measured to grade marketable pepper produced. The analysis of variance for fruit diameter has shown a highly significant (P<0.01) difference among irrigation levels. The largest fruit diameter (2.14 cm) was obtained from the plots that received full ETC, while the treatment that received 60% DI (1.23 cm) gave the smallest fruit diameter as compared to other treatments indicated in (Table 7 & Fig 13). 10, 20 and 30% DI showed an insignificant reduction in fruit diameter (P< 0.01), also 30, 40 and 50% DI showed the irrelevant differences. But full irrigation showed significant difference.

These finding shows, a large fruit diameter obtained from full irrigation was agreed with Yemane *et al.* [33] the largest diameter was recorded from full irrigation amount. Application of full irrigation via drip or furrow has resulted higher fruit weight and size [45]. On the other hand, the lowest diameter could record from irrigation levels treated with 60% DI. Higher percentage of large fruits, the significant difference observed among deficit irrigation levels on fruit diameter the present study therefore, reflection of variation in depth of water applied in respective deficit irrigation levels. The lowest fruit diameter got from 60% deficit levels indicated that slight imbalance in water supply can used drastic reductions in fruit growth. This finding was supported by Gobena *et al.* [46] reported that, deficit applied at bulb formation stage of onion resulted in significantly smaller bulb diameter than the control other treatments with 25% ETC deficit application. Bulb diameter was decreased with the irrigation water deficit at the development and bulb formation growth stages of onion [42].

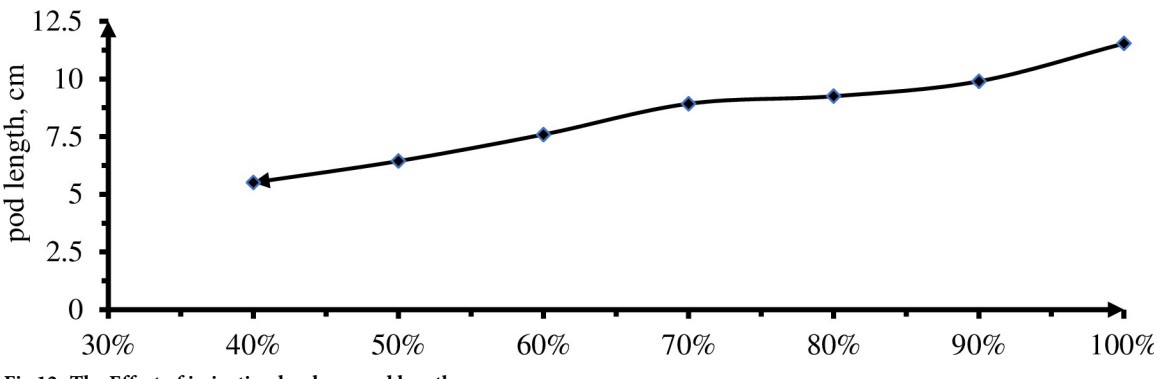

**Fig 12. The Effect of irrigation levels on pod length.**

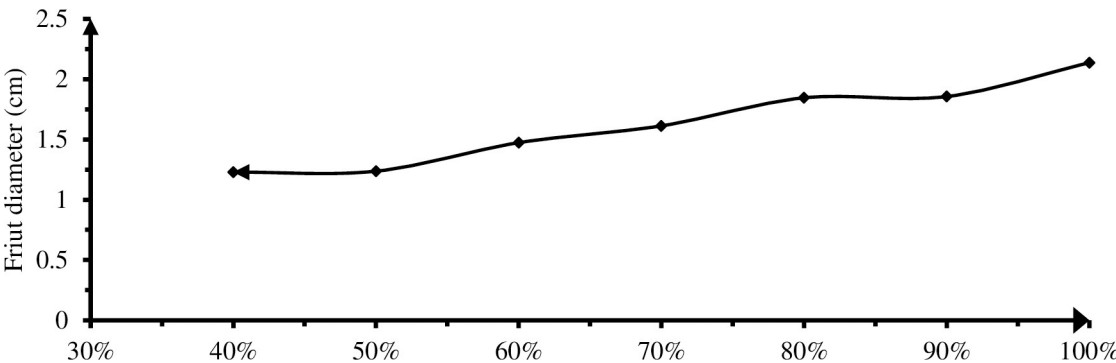

**Fig 13. Effect of irrigation levels on fruit diameter.**

**3.3.4 Fruits weight per plant.** Analysis of variance showed that attainments to fruit weight per plant of mareko fana pepper highly significant (P<0.01) difference were presented in (Table 7 & Fig 14).

The highest fruit weight per plant of 745.8 gm was observed from full ETC and the lowest with106.5g in 60% deficit level with a trivial difference with 50% DI. However, Fruit weight per plant showed there had a high significant difference in response to deficit irrigation application between each irrigation level. The result was in line with Metin *et al*. [44], when irrigation levels increased fruit weight per plant of mareko fana plant increased. The verdict also confirmed by Gireesh [23], irrigation levels were significantly influenced fruit weight per plant of chili. In addition, Alvaro *et al*. [32] the largest yield per plant was obtained under the highest irrigation levels.

**3.3.5 Marketable yield.** Analysis of variance on marketable yield showed that the effect of deficit irrigation levels performs a highly significant difference (P< 0.01) in all deficit levels on marketable yield (Table 9 & Fig 15). The highest fresh and dry marketable yield (13.60 and 6.43 t/ha) was obtained from 0% Dl while the lowest fresh and dry marketable yield (3.53 and 1.92 t/ha) was attained from 60% deficit levels.

The above result showed for each irrigation level, marketable yield decreased with increased in deficit levels. The trend implies marketable yield was lower as the soil moisture stress increase. This finding clearly shows that increased photosynthetic area in response to moisture availability significantly contributed to enhanced pepper productivity, possibly through the generation of additional assimilates. Increment of marketable yield as the irrigation amount

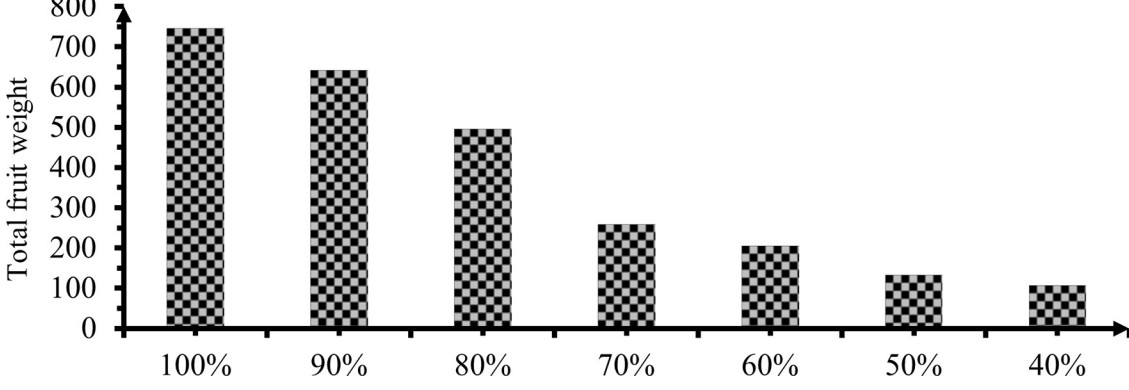

**Fig 14. Effect of irrigation levels on total fruit weight per plant.**

increased was similar to [47] indicated yield reduction related to increased soil moisture tension allowed continuing resulting loss of turgidity and termination of growth. Similarly, Patel and Rajput [48] also reported that water application without deficit at any growth stage got the highest marketable yield. As compared to full irrigation used of deficit irrigation water at 1/3 to 2/3 range during the development and middle stage did not affect pepper yield [13]. According to Tsegaye et al. [49], increased irrigation levels may result in more marketable onion bulbs due to an increase in the development of growth measures, resulting in quicker synthesis and movement of photosynthates from source to sink. Water stress during vegetative and fruit setting stage of bell pepper reduce leaf area, dry matter and yield [50].

The highest marketable yield reduction occurred when water stress was imposed at the plant phonological stages as well as completely growing season. In other way, while the stress level increased through plant phonological stages as well as completely growing season and the amount of water applied reduced then marketable yield reduced gradually [51]. Yet, all treatments were significant difference observed in marketable yield among each treatment. Due to significant difference in yield, it was not possible to locate the water stress threshold value (critical level of the area) of pepper.

**3.3.6 Unmarketable yield.** The analysis of variance showed that the effect of deficit irrigation levels resulted a significant difference ($P<0.01$) effect on unmarketable yield. Unmarketable dry yield shows significant difference at 0, 40 and 60% deficit but insignificant difference in other deficit levels ($P<0.01$) as shown in the above (Table 8 & Fig 16). The highest dry unmarketable yield was recorded from plants grown under 60% DI (0.35t/ha), followed by 50% DI (0.24 t/ha). While the lowest unmarketable yield was got from treatment received full irrigation (0.036 t/ha). ANOVA analysis of variance showed that the treatment that received 50, 40, 30 and 20% and also between10 and 0% deficit levels had not significantly difference but the other had significant differences.

The result revealed that, when deficit levels increased the small size, shirked, yellowish and white pod color produced more. Stressed pepper produced pod too early thus producing a high amount of unmarketable yield. This finding was supported by Yemane et al. [33] reported that the highest unmarketable yield 19.85 t/ha was recorded on irrigation level of 40% and the lowest value 1663.69 kg/ha was observed at 100% of water applied. Also, Edao and Quraishi [52] stated that, the maximum unmarketable pepper yield of 810.7 kg/ha was verified from the application of 0.5% ETc significantly difference compared to all other treatments. The

**Table 8. Effect of deficit irrigation levels on fresh and dry marketable, unmarketable and total yield of hot pepper at Alage in 2019/20 dry cropping season.**

| Treat. | FMY (t/ha) | FNMY(t/ha) | FTY(t/ha) | DMY (t/ha) | DNMY(t/ha) | DTY (t/ha) |
|---|---|---|---|---|---|---|
| 100% ET$_C$ | 13.60[g] | 0.20[a] | 13.80[f] | 6.430[g] | 0.037[a] | 6.54[f] |
| 90% ET$_C$ | 11.18[f] | 0.33[b] | 11.51[e] | 5.853[f] | 0.048[a] | 5.90[e] |
| 80% ET$_C$ | 9.46[e] | 0.38[c] | 9.84[d] | 5.45[e] | 0.113[ab] | 5.52[d] |
| 70% ET$_C$ | 9.07[d] | 5.545[d] | 9.63[d] | 4.69[d] | 0.114[ab] | 4.81[c] |
| 60% ET$_C$ | 6.83[c] | 1.06[e] | 7.89[c] | 3.54[c] | 0.191[b] | 3.73[b] |
| 50% ET$_C$ | 4.9[b] | 1.07[f] | 5.97[b] | 2.29[b] | 0.237[bc] | 2.53[a] |
| 40% ET$_C$ | 3.53[a] | 1.63[g] | 5.16[a] | 1.92[a] | 0.349[c] | 2.27[a] |
| s.e. | 1.484 | 0.1067 | 0.118 | 1.05 | 0.437 | 0.114 |
| c.v. | 1.8 | 1.6 | 1.6 | 3.0 | 13.5 | 3.1 |
| L.S.D. (1%) | 3.701 | 0.3260 | 3.601 | 0.325 | 0.1335 | 0.349 |

Where: FMY-fresh marketable yield, FUMY-fresh unmarketable yield, FTY-fresh total yield, DM-dry marketable yield, DNMY-dry unmarketable yield, (q/ha) and DTY-dry total yield CV-coefficient of variation and s.e-standard error.

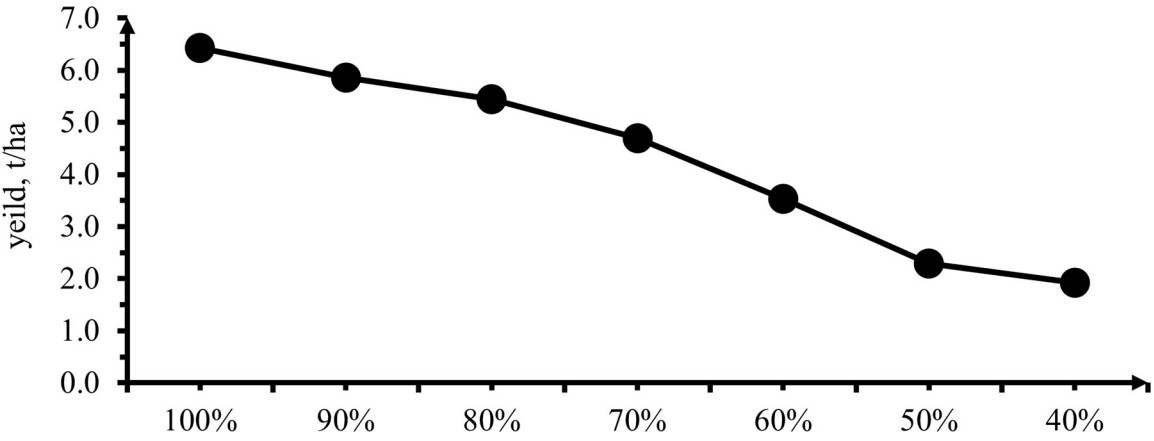

**Fig 15. The effect of irrigation levels on marketable yield per hectare.**

minimum unmarketable yields of 51.0 and 56.7 kg/ha were recorded from the application of 50% ETC. As Tadesse *et al.* [51] verdict showed increased soil moistures stress over the total growing season constantly and vegetative stages of plant results increased unmarketable yield.

**3.3.7 Total yield.** The sum of unmarketable and marketable pepper yields gives the total pepper yield. Total yield was highly influenced by deficit irrigation levels as shown in (Fig 17) below. ANOVA table showed that there was a highly significant ($P < 0.01$) difference in fresh and dry total yield of Mareko fana pepper on the effect of deficit irrigation levels (Table 8). At 50 and 60% on dry and 20 and 30% DI only showed insignificant differences ($P < 0.01$) but the significant differences on other deficit levels. Due to the significant difference in yield, there was not possible to locate the water stress threshold value of pepper but at 30% DI the yield reduction was nearly only one quarter (26.52%) of 0% deficit level by withholding 33.4% of water so, for water stress condition it is better to use 30% DI as maximum water stress value.

Maximum fresh and dry total yield (13.80 and 6.4 t/ha) from 0% deficit followed by crops that received 10% deficit levels (11.1 and 5.90 t/ha) respectively. Whereas, the minimum total fresh and dry yields (5.16 and 2.27 t/ha) were got from 60% deficit level. Total fruit yield was decreased by increasing deficit irrigation [37]. In addition, Adel *et al.* [30] testified that the highest total fruit yield was obtained from 100% followed by 85, 70, and 55% treatments in descending order. This drop in yield might be attributed to a number of variables, including a slower rate of photosynthesis, altered assimilate partitioning, or inadequate flag leaf formation. Wang and Bosland [53] yield reduction observed in the treatments that received less depth of

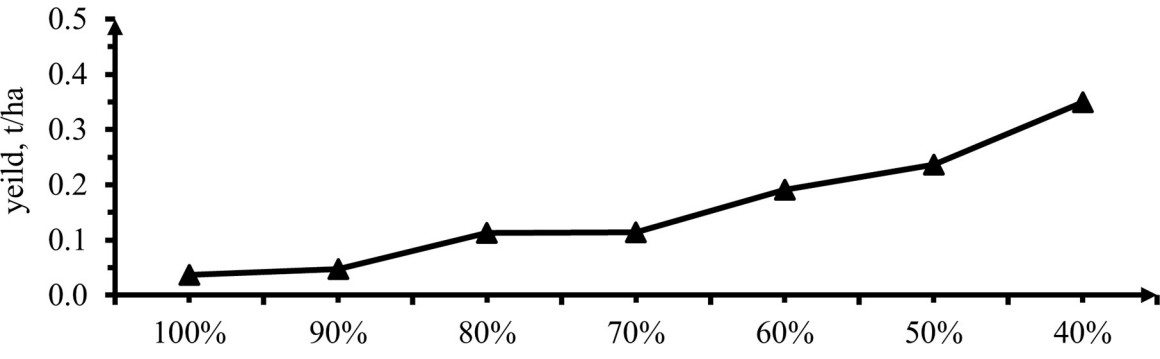

**Fig 16. The effect of irrigation levels on unmarketable dry yield.**

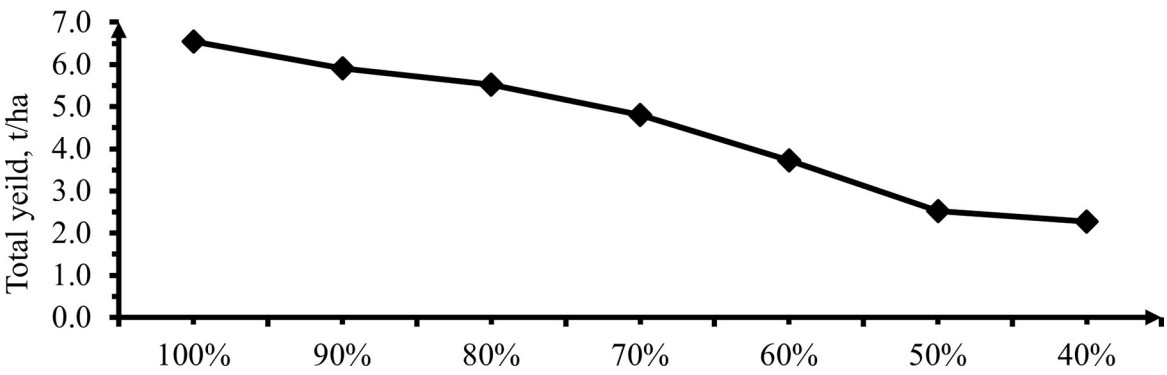

**Fig 17. Effect of irrigation levels on total yield per hectare.**

water per season resulted floral abortion, flower drop, immature fruit drop and reduction fruit number per plant and consequently total yield reduction due to water stress. According to Farooq *et al.* [54] total crop yield reduction occurred due to the drastically effect of drought conditions on various physiological and biochemical processes, including leaf respiration, chlorophyll content, gas exchange, leaf water content, and plant relative growth rate.

Water stress lowered assimilate transfer to the pod, resulting in a decrease in the number of endosperm cells, restricting the availability of assimilates or directly impacting the fruit synthesis process, consequently lowering fruit weight. Reduction in crop growth parameters, yield and yield component with reduced irrigation under different irrigation regimes and soil water depletions [51]. The general trend from this result observed yield of pepper increased with high depth of water supply and decreased with low depth of water supply under different irrigation levels throughout the whole growing season.

### 3.4 Drip irrigation water use efficiency

**3.4.1 Yield and water relationship.** The highest total yield of Mareko fana hot pepper 6.54 t/ha was obtained from 100% ETC. The treatment received 60% deficit levels throughout the growing season was produced 2.27 t/ha by saving 66.6% of water was the smallest total yield produced. The result exhibited that the highest yield reduction (65.32%) was observed under treatment received 60% DI. While the lowest yield reduction was observed from 20 and 10% DI.

Irrigate pepper at 10% of DI during the full growing season reduced the total yield by 9.81% and saved 11.1% of the irrigation water. Meanwhile, irrigation at 20, 30, 40 and 50% of the ETC reduced the total yield by 15.7, 26.52, 43.05 and 61.39% and saved water 21.6, 33.4, 44.5 and 55.5% of the irrigation water, respectively. Yet, Yield reduction was proportional to the amount of water applied under different deficit levels. Treatments that received 40, 50 and 60% DI had revealed the higher yield reduction as compared to10, 20 and 30% DI (Table 9). The same result by Adel *et al.* [30]. Meanwhile, irrigating at 30 and 20% of DI reduced the total yield by 26.52 and 15.7%, and saved 33.4 and 21.70% irrigation water respectively. Based on this result conditions when water was not a limiting factor for the crop under study 0% DI level enables to produce maximum yield but as indicated above 26.52% yield reduction was only obtained by withholding 33.4% of water applied for water stressed conditions at 30% DL.

This consistent decrease in yield with decreased water amount under different deficit levels is explained by the fact that when full crop water requirement was not met, water deficit in the plant causes stomata closure for the plant to save water, but at the expense of photosynthesis

**Table 9. Yield, water saved and relative yield reduction relationship under different deficit irrigation levels at Alage Central rift valley of Ethiopia.**

| Treatment | Crop water use (mm) | MY (t/ha) | UMY (t/ha) | TY (t/ha) | Relative yield reduction% | Water saved (mm) | % |
|---|---|---|---|---|---|---|---|
| 100% | 485.1 | 6.43 | 0.0365 | 6.54 | - | 0 | 0.0 |
| 90% | 436.5 | 5.853 | 0.0477 | 5.90 | 9.81 | 54 | 11.1 |
| 80% | 390.6 | 5.448 | 0.1127 | 5.52 | 15.70 | 105 | 21.6 |
| 70% | 339.3 | 4.694 | 0.1136 | 4.81 | 26.52 | 162 | 33.4 |
| 60% | 290.7 | 3.535 | 0.1909 | 3.73 | 43.05 | 216 | 44.5 |
| 50% | 243 | 2.29 | 0.2366 | 2.53 | 61.39 | 269 | 55.5 |
| 40% | 194.4 | 1.92 | 0.349 | 2.27 | 65.32 | 323 | 66.6 |

Where: MY- marketable yield, NMY- unmarketable yield and TY- total yield.

and biomass production [55]. Under field conditions, insufficient water supply could adversely affect the growth and yield of pepper. Beese et al. [56] stated that reducing water supply reduced chili pepper yield. When the water deficit occurred during the mid-stage the largest chili pepper yield reduction about 13–20% was recorded [57].

**3.4.2 Crop Water Use Efficiency (CWUE).** The CWUE in the study measures the effectiveness of the irrigation levels and drip irrigation methods in converting the total water applied to Mareko fana hot pepper yield. CWUE of Mareko fana hot pepper was (P< 0.05) affected by the effects of deficit irrigation levels as shown in (Table 10).

The highest crop water use efficiencies (14.22 kg/m3) were obtained from 60% deficit level, whereas the lowest crop water use efficiency (10.40 kg/m3) was obtained from full irrigation water levels. These results show a positive effect on irrigation deficit levels. CWUE was increased within the same water application method under different deficit levels. Hence, there was insignificant difference between 30, 40 and 50% DI, and other irrigation levels had significant different (P< 0.05). The result of this study indicated that, from each deficit level maximum crop water use efficiency was recorded from a plot treated with 60% deficit and also crop water use efficiency become lower as the deficit level reduced up to 0%. Therefore, when the water source was scarce, Mareko fana could be irrigated at the lowest water level (60% DI) taking economic conditions and minimum yields into consideration.

This result was confirmed by Arebu et al. [29], the maximum CWUE was obtained when 50% of the crop water requirement while the minimum was obtained from100% ETc. Yang

**Table 10. The effect of deficit irrigation levels on crop and irrigation water uses efficiencies of Mareko fana hot pepper at Alage in 2019/2020 dry season.**

| Treatment | CWUE (kg/ m$^3$) | IWUE (kg/ m$^3$) |
|---|---|---|
| 100% $ET_C$ | 10.40[b] | 9.36[a] |
| 90% $ET_C$ | 11.69[a] | 10.5[b] |
| 80% $ET_C$ | 12.80[c] | 11.54[c] |
| 70% $ET_C$ | 13.49[d] | 12.14[cd] |
| 60% $ET_C$ | 13.89[de] | 12.17[cd] |
| 50% $ET_C$ | 14.14[de] | 12.71[d] |
| 40% $ET_C$ | 14.22 [e] | 12.75[d] |
| Mean | 12.95 | 11.6 |
| L.S.D. (0.05) | 0.6593 | 0.82 |
| s.e. | 2.9 | 2.9 |
| CV (%) | 0.3026 | 0.27 |

*et al.* [57] also reported water deficit about 25–50% during the late stage is recommended for economic benefits and water productivity. The result was on contrary of Lelisa *et al.* [38], in which 70% irrigation water application and plastic mulch, the highest crop water uses efficiencies (8.48 kg/m$^3$) and the lowest crop water use efficiency (6.11 kg/m$^3$) was obtained from 50% water deficit and no mulch.

**3.4.3 Irrigation Water Use Efficiency (IWUE).** Irrigation water use efficiency is considered as relative pepper yield per unit of irrigation water used. IWUE revealed significant differences at 0, 20 and 60% Dl (P< 0.01) but insignificant difference on other levels (Table 10). The maximum IWUE (12.75 kg/m3) was verified from 60% DI followed by 12.71 kg/m$^3$ at 50% DI while, the minimum value of IWUE (9.36 kg/m3) was observed in full ETc levels. This exhibited that treatments with lower yield due to less water amount got higher IWUE.

The finding was confirmed by Arebu *et al.* [29] the highest IWUE (0.28 kg/m3) was obtained when 50% ETc while the lowest (0.22 kg/m3) was obtained when 100% ETc water was applied. Gebeyehu [58] said, IWUE related to the efficiencies showed that when irrigation water was limited, 70 and 55% deficit irrigation can be applied, increasing the water use efficiencies. When the deficit level increases IWUE and CWUE become increased. Additionally, Adel *et al.* [1] supposed, increase deficit irrigation resulted in insignificant increase in irrigation water use efficiency. In general, water use efficiency on fresh yield basis increased with more frequent irrigation application and WUE decreased with increasing irrigation levels. The result is also in line with Metin *et al.* [44] the gradually increasing water stress in the lower frequency irrigation treatments caused significant reductions in fruit yield, whereas higher frequency irrigation levels created a favorable soil water environment for pepper growth resulting in higher yields.

**3.4.4 Yield response factors (Ky).** Yield response factor (Ky) was derived from the relationship of relative yield reduction and evapotranspiration deficits for the whole growing period of hot pepper were given in Table 3. The result indicated that observed Ky for pepper fruit production ranged between 0.86 to 1.24. The lowest Ky (0.86) was observed under treatment that received 20% DI whereas, the highest Ky (1.24) was observed under the treatment received 50% deficit levels (Table 11). According to this result, irrigation level received 20% DI of full growth season was useful in saving irrigation water. Higher Ky values indicated the crop was at greater yield loss when the crop water requirements were not met. Doorenbos and Kassam [59] reported that pepper Ky would be greater than one, crop response was very sensitive to water deficit. Generally, it can be observed that Ky increasing with decreasing mareko fana yield and increasing in irrigation deficit levels.

The crop yield response factor (Ky) captures the essence of the complex linkages between production and water use by a crop. Crop yield response factor indicates a linear relationship

**Table 11. The yield response factor values for irrigation treatments.**

| Dl (%) | Yield (kg/ha) | Eta (mm) | $\frac{ETa}{ETm}$ | $\frac{Ya}{Ym}$ | $1 - \frac{ETa}{ETm}$ | $1 - \frac{Ya}{Ym}$ | Ky |
|---|---|---|---|---|---|---|---|
| 100% | 6543 | 485 | 1.00 | 1.00 | 0.00 | 0.00 | 0.0[a] |
| 90% | 5901 | 425 | 0.88 | 0.90 | 0.12 | 0.10 | 0.92[b] |
| 80% | 5516 | 388 | 0.80 | 0.84 | 0.20 | 0.16 | 0.86[b] |
| 70% | 4808 | 340 | 0.70 | 0.73 | 0.30 | 0.27 | 0.93[b] |
| 60% | 3726 | 291 | 0.60 | 0.57 | 0.40 | 0.43 | 1.10[bc] |
| 50% | 2526 | 243 | 0.50 | 0.39 | 0.50 | 0.61 | 1.24[c] |
| 40% | 2269 | 194 | 0.40 | 0.35 | 0.60 | 0.65 | 1.10[bc] |
| cv. | 13.1 | | | | | | |
| L.S.D. (1%) | 0.287 | | | | | | |

between the decrease in relative water consumption and the decrease in relative yield. Larger yield reductions when deficit increased because of stress which corresponded to Ky of this study shown on 40, 50 and 60% deficit levels. The crop with (ky >1) would suffer a greater yield loss than the crop with ky<1 as presented on deficit levels of 10, 20 and 30% deficit levels. Response factor greater than unity, indicated the expected relative yield reduction for given evapotranspiration deficit was equivalently greater than the relative decrease in evapotranspiration. Yield response factor exhibited significant difference only at 0 and 50% DI (P<0.01) but there was not significant different on other levels. Similar results had reported by Lelisa *et al.* [38].

## 3.5 Summary and conclusion

The results of this experiment showed that the water requirement of mareko fana pepper was low at initial stage, and it gradually increased in crop development stages, attaining a peak in the fruit establishing stage of pepper. Almost all parameters except flower and pod number per plant were recorded maximum under drip irrigation with full level compared to other treatments throughout the growing period. But the maximum number of flowers and pod numbers were recorded from 80% ETc levels.

Full Irrigation applications throughout the growing season was got minimum CWUE and IWUE but, 60% DI was got maximum compared with other irrigation levels (P<0.05). However, applying 60% DI throughout the growing season resulted maximum yield reduction but the relative water saved, CWUE and IWUE was maximum compared with other irrigation levels. The highest pepper Ky was obtained via 50% deficit throughout the growing season where the minimum yield reduction factor was observed at full irrigation level. Based on the yield response factor analysis 10, 20 and 30% deficit had minimum value whereas 40, 50 and 60% deficit levels have gotten greater than one (Ky>1) which achieved the maximum yield reduction (P<0.01).

Marketable yield showed a significant difference (P<0.01) in all deficit levels due to this; it was not possible to locate the water stress threshold value of pepper but, at 30% deficit level yield reduction was 26.52% nearly one-quarter of 0% deficit level by withholding 33.4% of water. Therefore, it concluded that when water is not a limiting factor for the crop under study, 0% deficit level enables to produce maximum yield but as indicated above 26.52% yield reduction is only obtained by withholding 33.4% of the water applied for water stress condition.

## 3.6. Recommendation

Based on the study and the results obtained on yield, yield components, and water productivity of pepper, the following important recommendations were made:

❖ To achieve maximum mareko fana pepper yield in areas where water was not scarce, applying full irrigation throughout the whole growing season under in line drip irrigation system was recommended.

❖ Even if difficult to get a specific threshold value of the area, based on yield reduction 30% deficit saved water 33.4% and produced 74.48% of yield. For water stressed conditions recommended to use 30% deficit for maximum deficit levels.

❖ Deficit intervals did not possible to locate the threshold value. This point may be located when similar studies at smaller deficit levels were employed.

❖ Similar studies should be carried out with different irrigation levels of moisture stress to ascertain conclusively the influence of the same study on yields and water productivity and to get the threshold value of the area.

## Acknowledgments

The authors would like to thank the Ministry of Agriculture for all the necessary and available facilities, equipments' and services delivered. Special thanks are forwarded to Alage ATVET College department of irrigation staff members who visited and assisted us during the study.

## Author Contributions

**Conceptualization:** Seid Mohammed.

**Formal analysis:** Arebu Hussen.

**Investigation:** Seid Mohammed.

**Methodology:** Seid Mohammed.

**Supervision:** Arebu Hussen.

**Validation:** Arebu Hussen.

**Writing – original draft:** Seid Mohammed.

**Writing – review & editing:** Arebu Hussen.

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
