## [Decision Letter · Decision Letter 0]

30 May 2022

PONE-D-22-10794Influence of Deficit Irrigation Levels on Agronomic Performance of Pepper ( Capsicum annuum L.) Under Drip at Alage, Central Rift Valley of EthiopiaPLOS ONE

Dear Dr. Arebu Hussen,

Thank you for submitting your manuscript to PLOS ONE. After careful consideration, we feel that it has merit but does not fully meet PLOS ONE’s publication criteria as it currently stands. Therefore, we invite you to submit a revised version of the manuscript that addresses the points raised during the review process.

We look forward to receiving your revised manuscript.

Kind regards,

Mario Licata, Ph.D.

Academic Editor

PLOS ONE

Journal Requirements:

5. We note that you have referenced (Ministry of Agriculture (MoA). 2012. Annual report on Irrigation development activities and its constraints. Addis Ababa, Ethiopia. Unpublished document module 5 (irrigation methods)) which has currently not yet been accepted for publication. Please remove this from your References and amend this to state in the body of your manuscript: (ie “Bewick et al. [Unpublished]”) as detailed online in our guide for authors

6. We note that Figure 1 in your submission contain map image which may be copyrighted. All PLOS content is published under the Creative Commons Attribution License (CC BY 4.0), which means that the manuscript, images, and Supporting Information files will be freely available online, and any third party is permitted to access, download, copy, distribute, and use these materials in any way, even commercially, with proper attribution. For these reasons, we cannot publish previously copyrighted maps or satellite images created using proprietary data, such as Google software (Google Maps, Street View, and Earth). For more information, see our copyright guidelines: http://journals.plos.org/plosone/s/licenses-and-copyright.

Additional Editor Comments:

Dear authors,

I have carefully read the manuscript. In my opinion the manuscript is interesting and enough original but it shows a lot of points of wealness. For example, the introduction is very poor and the results should be better presented and discussed. I suggest and recommend the authors to revise the manuscript and improve it. In particular, the authors should give more attention to physiological and chemical traits of pepper plants, increase the introduction adding recent references, improve the presentation and discussion of the results. My decision iis Major revision.

Kind regards

Reviewers' comments:

Reviewer's Responses to Questions

**Comments to the Author**

1. Is the manuscript technically sound, and do the data support the conclusions?

Reviewer #1: Yes

Reviewer #2: Partly

2. Has the statistical analysis been performed appropriately and rigorously? 

Reviewer #1: Yes

Reviewer #2: Yes

3. Have the authors made all data underlying the findings in their manuscript fully available?

Reviewer #1: Yes

Reviewer #2: Yes

4. Is the manuscript presented in an intelligible fashion and written in standard English?

Reviewer #1: Yes

Reviewer #2: No

5. Review Comments to the Author

Reviewer #1: 1- English writing needs more work (Editing)

2- In the introduction section- the objective of the study is not clear need more elaboration

3- The results and discussion very acceptable and clear

4- The last section, Conclusion, the authors must think what is next and what should be the future work through presenting their recommendations.

Reviewer #2: The manuscript titled ‘Influence of Deficit Irrigation Levels on Agronomic Performance of Pepper (Capsicum annuum L.) Under Drip at Alage, Central Rift Valley of Ethiopia’ submitted to PLOS ONE fit with the aim of the Journal. The authors conducted a trial to appraise the impact of deficit irrigation on yield, water productivity and to identify water stress threshold at different water deficit levels on pepper under drip irrigation methods.

In the present manuscript the authors focused their attention on the agronomic parameters, while very few attention was paid on physiological and biochemical traits that are, generally, affected by drought. The introduction section is very poor, the materials and methods section is enough detailed. The results and discussion section should be better presented and the results should be deeply discussed. A further statistical analysis (e.g. heat map) is strongly advised.

From my point of view, the manuscript should be improved in writing. Thus, although the manuscript is of high originality, I recommend the resubmission of the manuscript after the suggestions reported above.

6. PLOS authors have the option to publish the peer review history of their article (what does this mean?). If published, this will include your full peer review and any attached files.

Reviewer #1: **Yes: **Tala Qtaishat

Reviewer #2: No

---

## [Author Response · Author response to Decision Letter 0]

26 Sep 2022

Reviewer 1: I have incorporated all of your comments and suggestions into my revision. They were very important. Thank you very much for your help.

Reviewer 2: I have incorporated all of your suggestions into my revision. Thanks a lot for your help.

---

## [Decision Letter · Decision Letter 1]

4 Nov 2022

PONE-D-22-10794R1Influence of Deficit Irrigation Levels on Agronomic Performance of Pepper ( Capsicum annuum L.) Under Drip at Alage, Central Rift Valley of EthiopiaPLOS ONE

Dear Dr. Hussen,

Thank you for submitting your manuscript to PLOS ONE. After careful consideration, we feel that it has merit but does not fully meet PLOS ONE’s publication criteria as it currently stands. Therefore, we invite you to submit a revised version of the manuscript that addresses the points raised during the review process.

We look forward to receiving your revised manuscript.

Kind regards,

Mario Licata, Ph.D.

Academic Editor

PLOS ONE

Additional Editor Comments :

Dear authors

I have carefully read the comments of the two reviewers. One of them has accepted the manuscript in the present form, instead, the second one hase given a major revision. On the basis of these comments and after I have carefully read the manuscript, I think that the manuscript can be not accepted for publication in Plos One at this stage. I agree with the second reviewer due to fact that the authors have not looked for any qualitative traits. Therefore, the manuscript provides limited information. My decision is Major revision.

Furthermore, I suggested the authors to send the file with the main corrections (in red or with the revision program) next time.

Best regards

Reviewers' comments:

Reviewer's Responses to Questions

**Comments to the Author**

1. If the authors have adequately addressed your comments raised in a previous round of review and you feel that this manuscript is now acceptable for publication, you may indicate that here to bypass the “Comments to the Author” section, enter your conflict of interest statement in the “Confidential to Editor” section, and submit your "Accept" recommendation.

Reviewer #1: All comments have been addressed

Reviewer #2: (No Response)

2. Is the manuscript technically sound, and do the data support the conclusions?

Reviewer #1: Yes

Reviewer #2: Yes

3. Has the statistical analysis been performed appropriately and rigorously? 

Reviewer #1: Yes

Reviewer #2: Yes

4. Have the authors made all data underlying the findings in their manuscript fully available?

Reviewer #1: Yes

Reviewer #2: Yes

5. Is the manuscript presented in an intelligible fashion and written in standard English?

Reviewer #1: Yes

Reviewer #2: Yes

6. Review Comments to the Author

Reviewer #1: (No Response)

Reviewer #2: The authors addressed many points of criticism raised in the previous revision. However, since the authors did not look for any qualitative traits, the current work provides limited information. I suggest improving the discussions on the results.

7. PLOS authors have the option to publish the peer review history of their article (what does this mean?). If published, this will include your full peer review and any attached files.

Reviewer #1: No

Reviewer #2: No

---

## [Author Response · Author response to Decision Letter 1]

9 Dec 2022

Editor: Thanks very much for your comments and suggestions to make smart our manuscript. We made main corrections on all points raised in previous revision and ready for publication at this stage.

Reviewer 1: Thank you very much for accepted our manuscript in previous revision stage.

Reviewr 2: Thanks very much for your valuble comments and suggestions. We have incorporated you suggestions and comments into this second revision and make ready for publication.

---

## [Decision Letter · Decision Letter 2]

5 Jan 2023

Influence of Deficit Irrigation Levels on Agronomic Performance of Pepper ( Capsicum annuum L.) Under Drip at Alage, Central Rift Valley of Ethiopia

PONE-D-22-10794R2

Dear Dr. Arebu Hussen,

We’re pleased to inform you that your manuscript has been judged scientifically suitable for publication and will be formally accepted for publication once it meets all outstanding technical requirements.

Kind regards,

Mario Licata, Ph.D.

Academic Editor

PLOS ONE

Additional Editor Comments (optional):

Dear authors,

I am very satisfied by the work the authors carried out. They solved all problems the reviewers highlighted in the 1st-round review. The manuscipt has been improved following the suggestions of mine and reviewers.

No further comments.

Reviewers' comments:

Reviewer's Responses to Questions

**Comments to the Author**

1. If the authors have adequately addressed your comments raised in a previous round of review and you feel that this manuscript is now acceptable for publication, you may indicate that here to bypass the “Comments to the Author” section, enter your conflict of interest statement in the “Confidential to Editor” section, and submit your "Accept" recommendation.

Reviewer #2: All comments have been addressed

Reviewer #3: All comments have been addressed

2. Is the manuscript technically sound, and do the data support the conclusions?

Reviewer #2: Yes

Reviewer #3: Yes

3. Has the statistical analysis been performed appropriately and rigorously? 

Reviewer #2: Yes

Reviewer #3: Yes

4. Have the authors made all data underlying the findings in their manuscript fully available?

Reviewer #2: Yes

Reviewer #3: Yes

5. Is the manuscript presented in an intelligible fashion and written in standard English?

Reviewer #2: Yes

Reviewer #3: Yes

6. Review Comments to the Author

Reviewer #2: (No Response)

Reviewer #3: (No Response)

7. PLOS authors have the option to publish the peer review history of their article (what does this mean?). If published, this will include your full peer review and any attached files.

Reviewer #2: No

Reviewer #3: No

---

## [Editor Report · Acceptance letter]

12 Jan 2023

PONE-D-22-10794R2 

Influence of Deficit Irrigation Levels on Agronomic Performance of Pepper (*Capsicum annuum* L.) Under Drip at Alage, Central Rift Valley of Ethiopia 

Dear Dr. Hussen:

I'm pleased to inform you that your manuscript has been deemed suitable for publication in PLOS ONE. Congratulations! Your manuscript is now with our production department. 

Kind regards, 

on behalf of

Dr. Mario Licata 

Academic Editor

PLOS ONE